# Dye tracer aided investigation of xylem water transport velocity distributions

Stefan Seeger[1,2] and Markus Weiler[2]

[1]Soil Physics, Department of Crop Sciences, University of Göttingen
[2]Hydrology, Faculty of Environment and Natural Resources, University of Freiburg

**Correspondence:** Stefan Seeger (stefan.seeger@uni-goettingen.de)

**Abstract.** A vast majority of studies investigating the source depths in the soil of root water uptake with the help of water stable isotopes implicitly assumes that the isotopic signatures of root water uptake and xylem water are identical. In this study we show that this basic assumption is not necessarily valid, since water transport within a plant's xylem is not instantaneous. However, to our knowledge no study has yet tried to explicitly assess the distribution of water transport velocities within the xylem. With a dye tracer experiment we could visualize how the transport of water through the xylem happens at a wide range of velocities which are distributed unequally throughout the xylem. In an additional virtual experiment we could show that, due to the unequal distribution of transport velocities throughout the xylem, different sampling approaches of water stable isotopes might effectively lead to xylem water samples with different underlying age distributions.

## 1 Introduction

Water stable isotopic signatures ($\delta^2 H$ and $\delta^{18}O$) are a long established tool to study plant and root water uptake (Ehleringer and Dawson, 1992; Rothfuss and Javaux, 2017; Beyer et al., 2020) and transport. Since they are a part of the water molecule itself, they can be considered as ideal tracers to study soil-vegetation-atmosphere interactions. Additionally, in many regions of the world the isotopic composition of natural precipitation water exhibits a pronounced seasonality (Dansgaard, 1964; Rozanski et al., 1993). These naturally occurring fluctuations often lead to depth gradients of soil water isotopic signatures ($\delta_{soil}$), which can be utilized to infer the contribution of different soil depths to root water uptake (RWU).

Rothfuss and Javaux (2017) reviewed 156 papers that investigated RWU with the help of stable water isotopes. They could be grouped into three categories: 46% of them were categorized as linear inference methods, 50% of them were isotopic mixing models and a mere 4% (seven in total) were numerical methods with a more process based conceptualization of RWU. Irrespective of the chosen approach, all of the reviewed papers implicitly assumed an instantaneous equivalency between the isotopic signature in the soil and hence of RWU ($\delta_{\mathrm{RWU}}$) and the sampled isotopic signature of xylem water ($\delta_{\mathrm{xyl}}$).

### 1.1 Water transport within trees

While more recent scientific findings on the distribution of water transport velocities within tree xylem are either based on isotopic labeling (Kalma et al., 1998; James et al., 2003; Meinzer et al., 2006; Schwendenmann et al., 2010; Gaines et al., 2016)

or thermometric methods (Čermák et al., 2004; Gebauer et al., 2008), approaches based on dye tracers have been used long before. Harvey (1930) reported on his use of *Light Green SF* as a fast penetrating and non-toxic dye to trace the flow paths of water through plant xylem. Subsequently, dye tracers have been used in a number of studies (Baker and James, 1933; Müller, 1949; Mathiesen, 1951) to investigate the velocity of the transpiration stream in *Acer pseudoplatanus* (sycamore), *Fagus sylvatica* , *Fraxinus excelsior* (European ash) and *Betula spec.* (birch). More recently, the much less destructive thermometric methods (Marshall, 1958; Granier, 1985) became the dominant approach to investigate sap flux velocities and the role of dye tracers was mainly reduced to a supplementary mean to determine conducting sapwood areas (Dawson, 1998; McJannet et al., 2007; Lubczynski et al., 2017).

Based on the observations of Meinzer et al. (2006), who injected deuterium labeled water to the bases of mature trees and traced them to twigs in the crown and noticed considerable delays between tracer injection and tracer breakthrough at the crown, Berry et al. (2018) reasoned that there is an *inherent time lag between water entering the root and reaching the xylem at the point of sampling*.

In order to account for this time lag, Magh et al. (2020) temporally shifted their soil isotopic data by constant tree specific time lags for *Fagus sylvatica* (European beech) and *Abies alba* (silver fir) trees before inferring potential water uptake source depths with a linear mixing model in a Bayesian framework (Parnell et al., 2013). The tree specific time lags were determined by dividing $\delta_{xyl}$ sampling heights through observed sap flow velocities (determined by sensors based on the heat ratio method).

De Deurwaerder et al. (2020) went a step further and developed the more dynamic mechanistic *SWIFT* (Stable Water Isotopic Fluctuation within Trees) model which consists of two parts: the first part computes RWU similar to the Feddes et al. (1976) RWU model depending on the water potentials in different soil depths and within the roots to predict $\delta_{RWU}$ (de Jong van Lier et al., 2008). Driven by fluctuating sap flow velocities, only the advective propagation of this $\delta_{RWU}$ signature along the trunk xylem is then simulated by the second part of the *SWIFT* model in order to predict $\delta_{xyl}$ at specific sampling heights.

Knighton et al. (2020) have tested a similar piston flow representation and a well mixed representation of plant internal water storage for *Fagus americana* (American beech) and *Tsuga canadensis* (Canadian hemlock) within the ecohydrological model *EcH2O-iso* (Kuppel et al., 2018). Both approaches improved the agreement between simulated and measured $\delta_{xyl}$ compared to the default model version with no consideration of tree internal water transport or storage.

Seeger and Weiler (2021) presented field data comprising 10 weeks of daily in-situ measurements of $\delta_{soil}$ in six depths and $\delta_{xyl}$ at three trunk heights of *Fagus sylvatica*. They estimated $\delta_{RWU}$ with a Feddes type RWU model after Jarvis (1989) and achieved a generally good agreement between those estimated $\delta_{RWU}$ values and observed $\delta_{xyl}$ measurements for (artificially labelled) $^2H$ and (natural) $^{18}O$ signatures. However, they reported clear discrepancies between simulated $\delta_{RWU}$ and measured $\delta_{xyl}$ values in the wake of rapid $\delta_{RWU}$ changes. These discrepancies were largest for $\delta_{xyl}$ observations at 8m trunk height, but also clearly noticeable for $\delta_{xyl}$ observations at the base of the trunk. They utilized a convolution based transfer function approach to infer apparent flow path length distributions (FPLDs) between $\delta_{RWU}$ and $\delta_{xyl}$. This lead them to the conclusion, that the FPLD between root tips and trunk base is of a similar extent, but a different shape than the FPLD between trunk base and 8 m stem height. They did not manage to link the shape of the inferred transfer functions to any measurable characteristic of the studied trees.

**Table 1.** Combinations of sampling techniques (rows) and sampling domains (columns) that are discussed in this paper.

| | Q | $V_{\text{mobile}}$ | $V_{\text{total}}$ | Core |
|---|---|---|---|---|
| Physiochemical extraction | ✗ | ✗ | ✓ | ✓ |
| In-situ equilibration | ✗ | ✗ | ✗ | ✓ |
| Transpiration capturing | ✓ | ✗ | ✗ | ✗ |
| Pressure extraction | ✗ | ✓ | ✗ | ✗ |

Smith et al. (2022) adapted this transfer function approach for modelling the results of another in-situ isotope measurement study (Landgraf et al., 2022) on two *Salix* (Willow) trees with the ecohydrologic model *EcH2O-iso* (Kuppel et al., 2018). The parameters used to shape the transfer function have been determined via calibration and their final values were not reported or discussed. Compared to an instant mixing approach ($\delta_{\text{RWU}} = \delta_{\text{xyl}}$), the introduction of a transfer function between $\delta_{\text{RWU}}$ and $\delta_{\text{xyl}}$ did notably improve the agreement between simulated and observed $\delta_{\text{xyl}}$ values on a diurnal time scale, but not on a seasonal timescale.

## 1.2 Water pools captured by different sampling approaches

Apart from the proper representation of xylem water transport processes within models, another challenge for researchers working with $\delta_{\text{xyl}}$ data is the question of comparability between different approaches to sample $\delta_{\text{xyl}}$. Millar et al. (2022) have listed various combinations of sampling approaches and measurement techniques which may eventually lead to isotopic signature values that may enter a model. Within this study, we want to include some thoughts on how different $\delta_{\text{xyl}}$ sampling approaches may influence the measurement results due to xylem water transport processes.

Methods to sample plant water for stable water isotope analyses can be grouped into four categories (rows of Tab. 1). Under *Physiochemical extractions* we subsume solvent based extraction techniques (Thorburn et al., 1993) and the nowadays much more common cryogenic vacuum distillation, CVD (Koeniger et al., 2011). The possible sampled water pools (columns of Tab. 1) for all of these methods do overlap more or less, but they are not identical:

*Q*: flux of water that is flowing through the cross sectional area of a stem xylem segment within a certain time span. Methods that capture the transpiration of individual plants (Kalma et al., 1998; Dubbert et al., 2014; Volkmann et al., 2016a; Kulmatiski and Forero, 2021) are effectively sampling from this water pool.

$V_{\text{total}}$: all water that is contained within a whole stem xylem segment. Physiochemical extraction applied to entire branch segments (Thorburn and Ehleringer, 1995; Zuecco et al., 2022) captures this water pool.

$V_{\text{mobile}}$: all mobile water (i.e. water that is freely moving) that is contained within a whole stem xylem segment. This water pool can be sampled with a Scholander pressure chamber (Geißler et al., 2019; Magh et al., 2020; Zuecco et al., 2022) or a cavitron flow-rotor (Barbeta et al., 2022).

Core: all water contained within a xylem core sample. $\delta_{xyl}$ of such core samples can be obtained by physiochemical extraction (Dawson and Ehleringer, 1991), nowadays mostly CVD (Koeniger et al., 2011; Kahmen et al., 2021; Snelgrove et al., 2021; Fabiani et al., 2022). A similar domain is sampled by in-situ vapor equilibration approaches based on either probes (Volkmann et al., 2016b; Seeger and Weiler, 2021; Mennekes et al., 2021; Gessler et al., 2022) or boreholes through the whole stem (Marshall et al., 2020; Landgraf et al., 2022; Kühnhammer et al., 2022). Even though those boreholes are going through the whole stem, Marshall et al. (2020) presented some model computations that suggest that this method effectively samples only from the outer most few centimeters of the xylem.

## 1.3 Study objectives

Since water has to be transported from one point in the xylem to another and this transport is by no means instantaneous, we should not assume that $\delta_{RWU}$ is directly reflected by measurements of $\delta_{xyl}$ at the same point in time. While several isotopic labeling studies have shown that $\delta_{xyl}$ between two different points in the xylem are different due to transformative effects of xylem water transport, no study has yet tried to explicitly assess the distribution of tracer transport velocities within the xylem. The primary objective of this study was to test whether an inexpensive dye tracer approach can be used to chart the distribution of water transport velocities within tree xylem. The secondary objective of this study was to compare the underlying water transport velocity distributions associated to different $\delta_{xyl}$ sampling approaches.

## 2 Methods

### 2.1 Dye tracer experiment

#### 2.1.1 Experimental setup

In order to visualize the distribution of tracer transport velocities within tree xylem, we conducted a dye tracer experiment whose setup is illustrated in Fig. 1 and which will be described in the following section.

We selected a 95 cm long stem segment (diameter of 3.5 cm) of a young *Salix* (willow) tree. The stem segment was harvested shortly before leaf-out in April 2022 from a garden on drained Gleysol in the municipality of Ihringen, Southwestern Germany (48°3′N 7°39′E, 190 m a.s.l.). There is no inherent importance of the species, size or provenience of the selected specimen, it only serves as an example to test the presented method.

Between sampling the stem segment and the start of the experiment (approximately 24 hours), we wrapped both ends of the stem segment in wet towels covered by plastic bags to avoid cavitation. Shortly before the start of the experiment, we cut another 5 cm of both ends of the stem segment. Subsequently, the stem segment was fixed to a rig, upside-down compared to its original orientation. A large funnel was placed on top of it. The base of the stem segment was connected to the outlet of the funnel with a piece of bicycle tubing and hose clamps. Below the stem segment, we placed a beaker that collected all water dripping out of the stem segment (Fig. 1a). Initially, we filled the funnel with tap water and waited for a few minutes to ensure that we have a steady flow of water through the stem segment. Then, we emptied the funnel and the beaker and filled the former

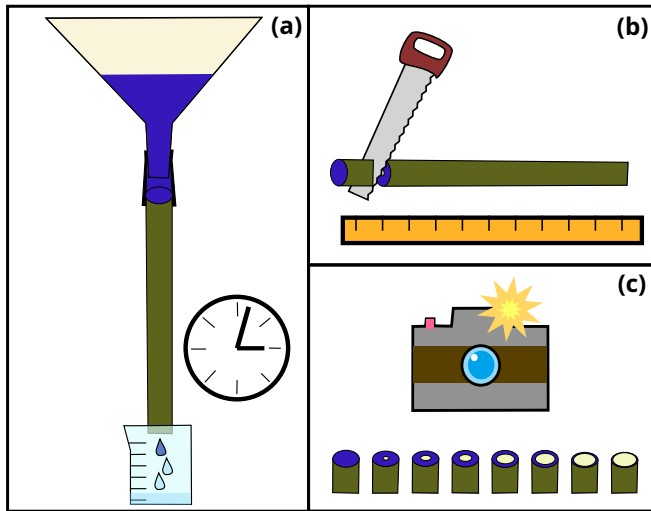

**Figure 1.** Sketch to illustrate the steps of the dye tracer experiment: **(a)** Application of dye tracer to stem segment; **(b)** Cutting the stem into smaller sub-segments; **(c)** Capturing images of cut surfaces for each sub-segment

one with 4 litres of dye infused tap water (4g/L Brilliant blue FCF). As soon as the first visible dyed drop of water left the
stem segment, we removed the stem segment from the rig and placed it horizontally to stop the gravitation driven water flow.
Then we used a band-saw to cut the stem segments into approximately 5 cm long sub-segments (Fig. 1b). After measuring the
segments' diameters and lengths, they were placed next to each other and photographed (Fig. 1c).

### 2.1.2    Data processing

We used the image processing software ImageJ (Schneider et al., 2012) to manually crop each of the stem sub-segments out of
the captured image and stored them in separate image files (Fig.2a). Subsequently, we used ImageJ to manually create masks
that indicate the xylem (excluding image background, bark and phloem) of each sub-segment and stored them in another set of
files (semi-transparent overlay in Fig.2b).

    With the cropped image files of each stem sub-segment and the respective masks, we automated the further data processing
within the R-software environment (RCoreTeam, 2022) were we used the package *readbitmap* (Jefferis, 2018) to load the image
files and their respective xylem masks. Apart from excluding non xylem pixels from the automated analysis, the xylem masks
were also used to determine the distance of each xylem pixel to the nearest non xylem pixel. These xylem border distances
were then used to assign each pixel to one of six xylem depth classes, equally spaced from the outer xylem border (see colored
lines in Fig. 2c). We decided to use six xylem depth classes, since we estimated the age of the stem segment to six years, even
though we could not distinguish any year rings.
In order to automatically decide whether a xylem pixel should be counted as stained from the blue dye tracer, we adapted
the formula used to determine the NDVI (normalized difference vegetation index (Tucker, 1979; Hayes, 1985)) and computed

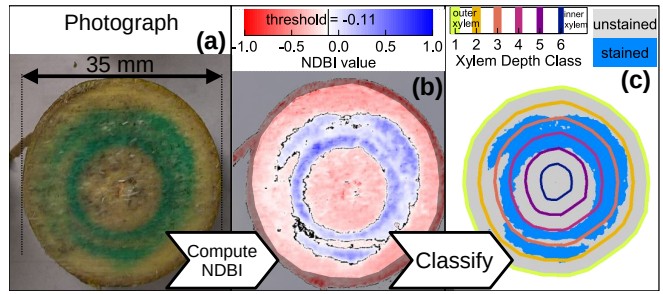

**Figure 2.** Exemplary images for the processing of cut surface #15. **(a)** shows the original photograph, **(b)** shows the computed NDBI values overlain by a mask that excludes everything but the xylem and **(c)** shows the detected blue stained area (light blue pixels) within six xylem depth classes.

a "normalized difference blueness index" (NDBI), based on the blue ($B$) and red ($R$) RGB values of each pixel:

$$\text{NDBI} = \frac{B - R}{B + R} \tag{1}$$

Based on our own visual comparisons, we classified pixels with an NDBI $< -0.11$ as unstained and those with NDBI $>= -0.11$ as stained (Fig.2 b & c). Based on this classification, we determined the fractions of blue stained area ($F$) for cut surface $n$ and xylem depth class $d$ as:

$$F_{n,d} = S_{n,d}/A_n \tag{2}$$

where $A_n$ is the number of all pixels in the respective cut surface's xylem mask and $S_{n,d}$ is the number of stained pixels within xylem depth class $d$. Subsequently we determined the differences in stained area fractions between adjacent cut surfaces for each depth class with:

$$f_{n,d} = F_{n-1,d} - F_{n,d} \tag{3}$$

where $F_{n-1,d}$ belongs to the cut surface closer to the stem base. Finally, we computed the required velocities $u_n$ to reach each cut surface within the tracer application time $t$ with the simple equation:

$$u_n = s_n/t \tag{4}$$

where $s_n$ is the distance between the stem base and the original position of the $n^{\text{th}}$ cut surface and $t$ is the duration of the dye tracer infiltration. The combination of $f_{n,d}$ and $u_n$ represents velocity density distributions for different xylem depths whose resolution is given by the number of cut surfaces. The mean tracer transport velocity within each depth class $d$ could then be computed as:

$$\overline{u_d} = \sum_{n=1}^{N} (f_{n,d} \times u_n) / \sum_{n=1}^{N} f_{n,d} \tag{5}$$

 where $N$ is the total number off all cut surfaces. The overall transport velocity $\overline{u}$ is computed similarly as an area weighted mean of all $\overline{u_n}$ values.

Complementarily, we also computed $q$, the volumetric flux of water through the whole stem segment:

$$q = \frac{Q}{A} = \frac{V}{At} \tag{6}$$

where $A$ a is the cross sectional area of the stem segment and $Q$ the discharge, which is equal to $V$ (the volume of water which
 dripped into the beaker) divided by the respective duration $t$.

## 2.2 Velocity distributions of water samples captured by different sampling approaches

Water samples taken from a xylem segment may capture different water velocity distributions depending on the applied sampling approach. In a virtual experiment we first generated a hypothetical xylem segment with a known idealized distribution of water transport velocities and then we compared the effective velocity distributions that would be captured by five different
 sampling approaches.

### 2.2.1 Generation of virtual xylem segment with idealized transport velocity distribution

For our virtual experiment, we assumed an idealized cross section of a stem segment (depicted in Fig. 3a). This cross section is characterized by an idealized distribution of tracer transport velocities similar to the observations we made in the dye tracer experiment that is presented in this study. We arbitrarily conceptualized the xylem segment with $D \; (= 12)$ equally spaced
 xylem depth classes. Possible transport velocities were discretized in $N$ steps reaching from $u_0 = 0$ to $u_N = u_{max}$. Now the frequency density matrix $f^*_{n,d}$ (with $n$ in $[0, ..., N]$ and $d$ in $[1, ..., D]$) can be used to describe the frequencies of certain velocities at certain xylem depth classes. This depth class specific velocity density distribution described by $f^*_{n,d}$ can be transformed into the xylem wide velocity density distribution $f_{n,d}$ (see Eq. 3), by weighing the densities of each depth class with the respective depth class' area fraction of the total cross sectional area.

 In order to fill $f^*_{n,d}$, we iterated over all xylem depth classes ($d$ between 1 and 12), and assigned the frequencies of certain velocities by computing a Gaussian density distribution with a standard deviation of $\sigma = 0.08\overline{u_{max}}$ around the respective depth class' mean velocity $\overline{u}_d$. Based on the results of our dye tracer experiment, we set the mean transport velocity at the outer xylem $\overline{u_1}$ to a fairly high value, increased it even further for the next two depth classes, so that $\overline{u}_3 = \overline{u_{max}}$. From there on, we decreased the assigned velocities with increasing xylem depth. The two innermost xylem classes were defined to have a transport velocity
 of zero. Similar radial velocity distributions based on sap flux measurements have been reported by Lüttschwager and Remus (2007) and Čermák et al. (2008). Finally, we added an immobile water fraction $\phi_i = 0.1$ to all depth classes where $\overline{u}_d > 0$ (all but the two innermost depth classes) according to the following equation:

$$f^*_{0,d} = \frac{\sum_{n=1}^{N} f^*_{n,d}}{\phi_i^{-1} - 1} \tag{7}$$

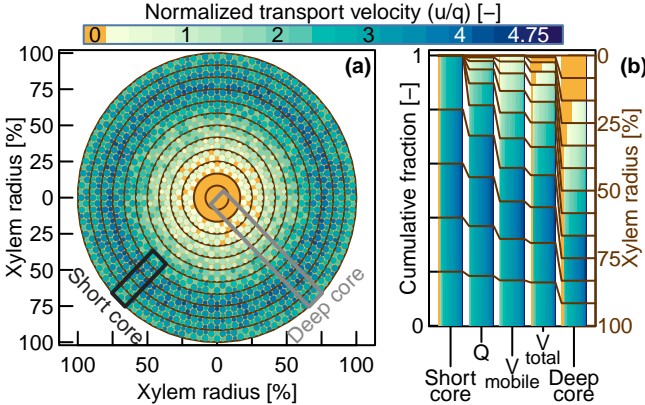

**Figure 3. (a)** Conceptual velocity distribution within an idealized xylem cross section. The mobile water velocity distributions that were assumed in our virtual experiment are represented by four classes of coloured circles for each xylem depth class. Theses four classes of circles represent the mean velocities between the 15%, 50%, 85% and 100% percentiles of the assumed velocity distributions of the respective xylem depth classes. Immobile water, depicted in orange, is present in each xylem depth class, since we assumed a constant fraction of cell bound immobile water. **(b)** Compositions of effective velocity distributions associated to hypothetical water samples acquired by different sampling approaches (see Sec. 2.2.2).

This immobile water fraction is meant to account for water that might influence $\delta_{xyl}$ measurements while it is not actively
participating in the transport of water through the xylem. The value of 10% was chosen rather arbitrarily since we could not find any study that reported the ratio of mobile to immobile water within tree xylem.

### 2.2.2  Virtual sampling

At that point, $f_{n,d}^*$ represented the information about the frequencies of specific velocities at specific depth classes within the xylem. Different sampling approaches will capture different parts and/or proportions of this original $f_{n,d}^*$. For a virtual method
inter-comparison we considered the four approaches described in Sec. 1.2, but split the *Core* case into *Deep core* (spanning the whole xylem depth) and *Short core* (limited to the outer five xylem depth classes as depicted in Fig.3a):

- $Q$: $f_{n,d}^*$ is weighted by $A_d$ (area of each xylem depth class $d$) and $u_n$ (velocity of each velocity class $n$).

- $V_{\text{total}}$: $f_{n,d}^*$ is weighted by $A_d$.

- $V_{\text{mobile}}$: $f_{n,d}^*$ (with $n >= 1$) is weighted by $A_d$. $f_{0,d}^*$ (the fraction of immobile water across all depth classes) is set to 0.

- Deep core: $f_{n,d}^*$ is taken as it is.

- Short core: $f_{n,d}^*$ is taken as it is, but all values where $d > 5$ are set to 0.

# 3  Results

## 3.1  Transport velocities within the xylem

With the timer starting at the first application of the dye tracer, it took 42 minutes until the first drops containing a visually detectable amount of blue dye tracer dropped into the beaker placed under the stem segment. Combined with the 86 $cm$ total stem length, this implies a maximum tracer transport velocity of 29.1 $md^{-1}$. Within that time, 140 mL of water were collected in the beaker. With a xylem diameter of 32.5 $mm$ (measured at the thinner end of the stem segment, 38 $mm$ measured at the thicker end), the cross sectional area amounted to 8.3 $cm^2$. Consequently, the flux of water through the stem segment was 5.7 $md^{-1}$ (Eq. 6).

Cut surface #18, farthest away from the tracer injection point, was slightly stained, but with an NDBI threshold of -0.11 our processing routine could not identify any stained areas for this cut-surface. The visibly stained areas in cut-surface #18 could be correctly classified with lower NDBI threshold values, but at the cost of obvious false positives at other cut surfaces. For the sake of identical treatment, we eventually decided to analyze all cut-surface images with an NDBI threshold of -0.11, well knowing that this may lead to a slight underestimation of the maximum tracer transport velocities. According to the sizes of the cropped images for each cut-surface and the actual dimensions of the respective stem segments, the resolution of our analysis was about 14x14 pixels per $mm^2$.

Figure 4a shows digitized images of the stem segment cut-surfaces. Pixels with NDBI values above the threshold of -0.11 are depicted blue. The colored contour lines (yellow to dark blue) indicate the outer extents of six automatically delineated xylem depth classes. Figure 4b shows the fractions of dye stained area grouped by the respective xylem depth classes. Starting with close to 100% dye stained area at the first cut surface (which was directly exposed to the dye tracer at the outlet of the funnel), the stained area fraction diminished quickest at the innermost xylem depth class and the second innermost depth class. For the cut surfaces #1-6, the fraction of stained area of the four outer depth classes remained fairly constant at around 95%. After that, the stained area fractions decreased for all depth classes. The outermost depth class showed no more dye stains after 70 cm (cut surface #15). The second outermost depth class was most persistent, still showing considerable dye traces up to the penultimate cut surface #17.

The dye tracer transport velocity distribution for the whole stem is represented by the stacked colored areas in Fig. 5a. With a value of 5.7 $md^{-1}$, the xylem water flux velocity $q$ (computed with Eq. 6) was less than a third of the mean tracer transport velocity $\bar{u}$ with a value of 18.6 $md^{-1}$ and about one fifth of the highest tracer velocity of 28 $md^{-1}$, which was observed at the 3rd, 4th and 5th xylem depth class.

Figure 5b depicts the individual tracer transport velocity distributions for the six xylem depth classes. The velocity distributions for the outer three depth classes resembled a normal distribution, while the fourth and fifth depth classes featured skewed velocity distributions. The tracer velocity distribution of the inner most xylem depth class had a clear peak at 0 $m^1d^{-1}$. Complimentary, Fig. 5c shows the tracer velocity distributions plotted against the xylem depth. Tracer velocities were increasing from the xylem center to the outer xylem, peaking at around 60% of the xylem radius and then slightly decreased towards the outer xylem.

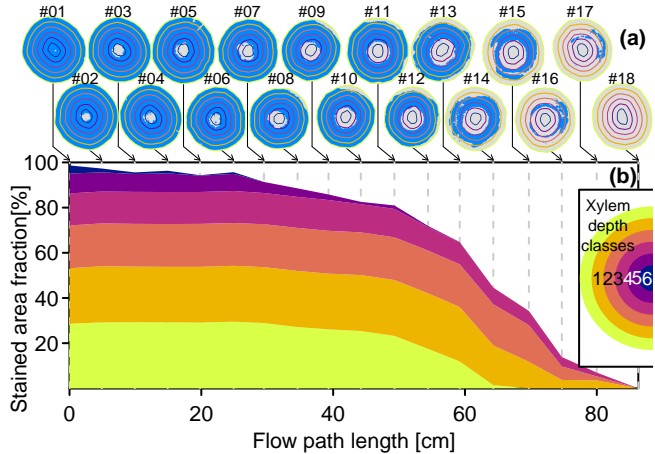

**Figure 4. (a)** Digitized cut surface images with detected stained areas depicted in blue, overlain with outer extents of xylem depth classes (yellow to dark blue lines). **(b)** Cumulated stained area fractions by xylem depth class derived from the processed cut surfaces images shown in (a)

### 3.2 Transport velocity distributions captured by different sampling approaches

The transport velocity distributions (TVDs) of water samples captured by different hypothetical sampling approaches for one and the same xylem segment are shown in Fig. 6a-e. For reference purposes, the TVD of the xylem water flow (TVD$_Q$) was plotted as a grey area into each of the TVD plots shown in Fig 6 and all velocities were given as multiples of the volumetric flux $q$. While the mean transport velocities ($\overline{u}$, dashed vertical black lines in Fig. 6) ranged from $1.7q$ (Fig. 6d) to $3q$ (Fig. 6a), the highest captured transport velocity was the same for all sampling approaches. The TVD for the sampling of mobile water (Fig. 6b) was most similar to TVD$_Q$, even though the outer xylem was slightly under- and the inner xylem slightly over represented. The short core's TVD (Fig. 6e), which completely excluded the inner xylem, was surprisingly similar to TVD$_Q$, while the deep core's TVD (Fig. 6d) showed the least resemblance to TVD$_Q$ due to a strong over representation of the much less mobile inner xylem water. While the TVDs of the xylem water flow and the mobile water volume (Fig.6a&b) were not influenced by immobile water, the other three cases (Fig.6c-e) contained immobile water fractions between 10 and 29%. The latter maximum value was reached from the deep core sampling scenario, which sampled all xylem depths classes to equal parts and thereby incorporated a disproportionally high fraction of the innermost xylem, which is not actively taking part in the water transport.

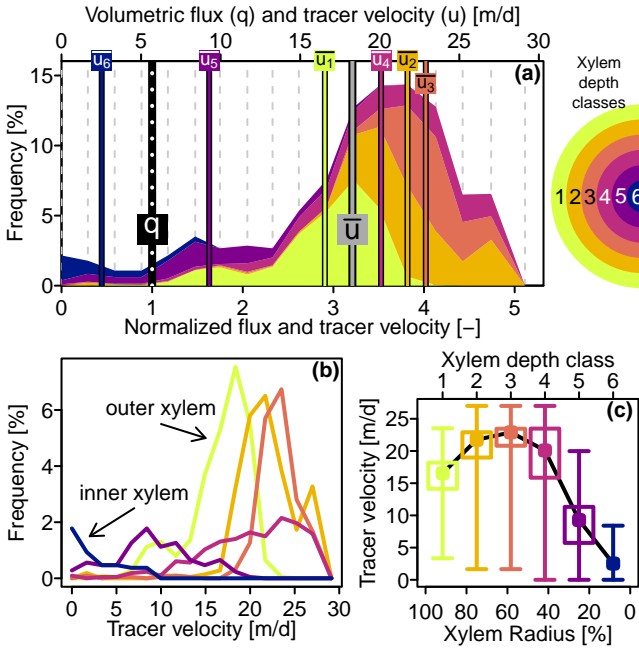

**Figure 5. (a)** Stacked colored areas show the contributions of six xylem depth classes to the overall tracer transport velocity distribution. Vertical bars indicate mean transport velocities per depth class ($\overline{u_1}$ to $\overline{u_6}$), overall mean transport velocity $\overline{u}$ and volumetric flux $q$. The x-Axis on the top shows the observed velocities, whereas the bottom x-Axis shows the same values normalized by the volumetric flux. **(b)** Tracer transport velocity distributions for the individual xylem depth classes (same color code as in (a)). **(c)** Box plots of tracer transport velocity distributions plotted against xylem depth – whiskers indicate depth classes' absolute ranges, boxes the 25-75 % quantile ranges and points the mean values.

## 4 Discussion

### 4.1 Dye tracer experiment

In order to make full use of the potential of the proposed method to infer tracer velocity distributions within the xylem, several aspects of the lab experiment introduced in this study could be improved.

The general purpose band-saw that we used to cut the stem into sub-segments produced rather rough cut surfaces and occasionally left some grime. At the time of the image capturing, some of the sub-segments' cut-surfaces were partially covered by saw dust. This could be improved by using a cleaner and finer wood saw and more thorough post-treatment of the cut-surfaces (e.g. blasting them with pressurized air or sanding).

Another consideration would be a more controlled way to take the images. We placed all 18 sub-segments next to each other and photographed all of them at once under ambient light. This lead to a) optical distortion of the segments placed off the center of the image and b) non-uniform lighting conditions. Photographing each segment individually directly from above and at a constant distance could greatly reduce optical distortions, and controlled lighting combined with reference color

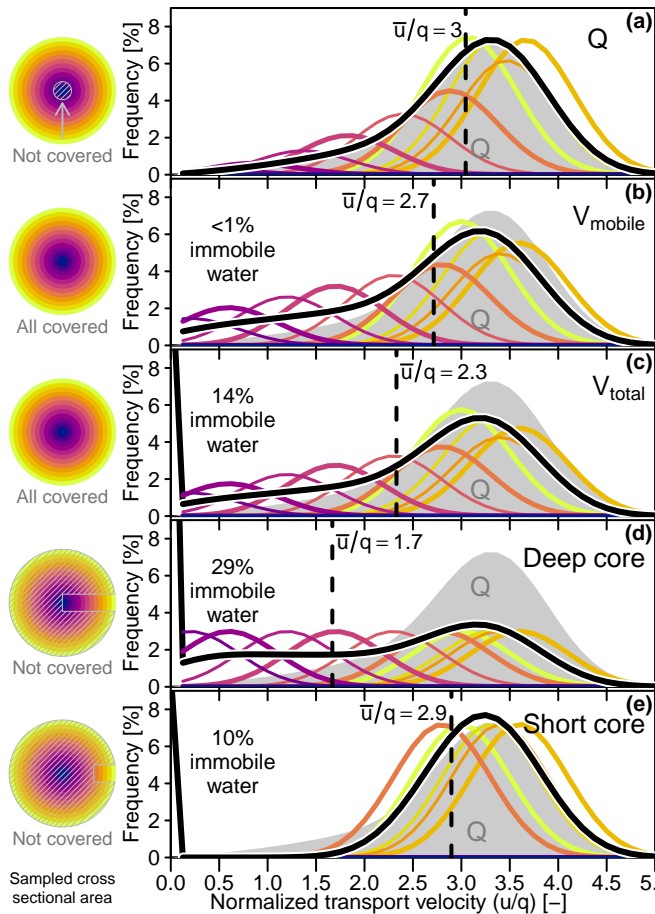

**Figure 6.** Left column: cross sectional areas that contribute to samples obtained with different approaches. Right column: water transport velocity distributions captured by a sample (thick black line) and the contributions of different xylem depths (colored lines, scaled by 400%) for different sampling approaches. Vertical dashed lines indicate the respective mean transport velocities of each sampling approach. All velocities are normalized by the flux velocity $q$. **(a)** Actual throughflow (Q), **(b)** Mobile water volume ($V_{mobile}$), **(c)** Total water volume ($V_{total}$), **(d)** Deep core, **(e)** Short core.

swatches would enable a more quantitative evaluation of the dye patterns by accounting for the staining intensity instead of just classifying into stained/unstained areas. Individual images for each cut surface could also easily improve the spatial resolution of the further analysis to well beyond our 14x14 pixels per $mm^2$. On top of that, it might be much easier to automatically process these images and it might eliminate the need for the manual steps of cropping the individual segments and creating xylem masks. In the best case, it might even be possible to automate the identification of single year rings and get rid of the rather arbitrary depth class zoning.

However, there is a certain limit to the time that can be spent to post-process and photograph the stem segments. Twelve hours of storage of the sub-segments at room temperature (exposed to the atmosphere) lead to evaporation from the cut-surface

which caused additional water and dye tracer movement, completely confounding the patterns we were interested in. We managed to capture the patterns quickly (all 18 sub-segments were cut and captured within one image in about 10 minutes), but more thorough post-processing and imaging might require special measures to reduce post-cut evaporation. Umebayashi et al. (2007) reported that immersion in liquid nitrogen and subsequent freeze drying of fresh cut xylem conserved the dye distribution at cutting time and allowed for more elaborate post processing steps like slicing with a microtome.

## 4.2 Tracer velocity distributions within the xylem

Even though a comparison between the highest transport velocities observed in our dye tracer experiment and those (implicitly) reported in the studies of Baker and James (1933), Müller (1949) and Mathiesen (1951) would be possible, it would be problematic. In our study we applied a nearly constant positive water potential (starting the experiment with 4 L of water in the funnel atop the stem base and ending it with 3.86 L, while the aforementioned studies all worked with the not further specified natural transpiration stream which can be expected to vary during and differ between single experiments conducted at different times and locations.

Instead of looking at specific velocities, it might be more helpful, to focus on the observed ratio of maximum transport velocity to the observed volumetric flux. Our dye tracing experiment suggested that the fastest tracer transport velocities ($u_{\mathrm{max}}$) were about 5 times as fast than the mean volumetric flux $q$. Other dye tracer studies did not report the respective volumetric fluxes, but several studies that traced the propagation of a $D_2O$ pulse injected to the base of the trunk of trees up to the leaves (Meinzer et al., 2006; Schwendenmann et al., 2010; Gaines et al., 2016) have reported maximum tracer velocities to be about 4 to 16 times faster than mean sap flux velocities. Those velocities have been shown vary considerably between different species and also between differently aged individuals of the same species. Seeger and Weiler (2021) observed a deuterium labeling pulse passing at two in-situ xylem water isotope probes installed at 0.1 and 8 m height also observed that maximum tracer transport velocities and sap flux velocities differ by a factor of 5.5. Mennekes et al. (2021) reported tracer velocities much closer to the observed sap flux velocities, but the investigated trees were comparatively small and the distances may have been too short in relation to $\delta_{xyl}$ observation frequencies to actually capture the fastest fraction of the tracer signal.

Although the *Brilliant Blue FCF* dye tracer used in this study has a larger molecular structure than the water itself, its application within a cut stem led to observations in good agreement with isotope based studies. Nevertheless, a direct comparison between the breakthrough curves of stable water isotopes and different dye tracers would be interesting and could increase the confidence in the validity of the approach of this study.

With regard to the depth distribution of tracer transport velocities, the study of Müller (1949) reports the highest transport velocities to occur in the outermost two year rings of *Fagus sylvatica* and a steady decline of transport velocities towards the inner xylem, with still detectable dye tracer transport up to the 24[th] youngest year ring of a 40 year old tree. Gebauer et al. (2008), who used thermoelectric sap flux sensors at different xylem depths to infer radial velocity distributions for sap flux, reported a similar pattern (highest velocities in the outermost xylem) for the sap flux velocities of *Fagus sylvatica* (European beech). But for *Acer pseudoplatanus* (sycamore maple), *Acer campestre* (field maple), *Carpinus betulus* (hornbeam) and *Tilia spec.* (linden) they observed a pattern similar to the one observed in this study, with a velocity maximum close to

the outer xylem but a slight velocity drop towards the outermost xylem. Lüttschwager and Remus (2007) and Čermák et al. (2008) reported similar radial distributions (with a maximum slightly below the outermost xylem) for sap flux velocities of *Fagus sylvatica* and *Pinus sylvestris* (Scots pine), respectively. At this point we have to emphasize, that in this study we were looking at dye tracer transport velocities, while an overwhelming majority of recent studies is reporting radial sap flux velocity distributions based on an entirely different measurement principle. Tracer transport velocities seem to be generally higher than sap flux velocities, which comes as no surprise, since the former focus on the moving water alone, whilst the latter are referring to the volumetric flux, which by definition will always be lower for a porous medium. Those basic differences aside, transport velocities and sap flux velocities are tightly linked to each other and share similar radial distributions. Within this study, we presented a methodology that can resolve those radial distributions with a much higher spatial resolution than current sap flux measurement devices.

It should be noted that with the presented experimental setup we were solely looking at one specimen of one tree species at one particular boundary condition provided by a nearly constant and uniform water potential applied to the base of the stem segment. Čermák et al. (2008) observed changes in the sap flux velocity distribution over xylem depth of *Pinus sylvestris* with decreasing top soil water availability. This lead to a notable decrease of sap flux velocity in the outer xylem (connected to rather shallow lateral roots) while sap flux in the inner xylem (connected to deeper roots) was maintained at a level close to the pre-drought conditions. Consequently, it might be better to interpret the results of our dye tracing experiment as a conductivity distribution instead of a velocity distribution. Only at uniform water potentials across the whole radial xylem depth range the two distributions can be expected to be congruent.

The dye tracer experiment presented in this study was performed on a specimen of willow, which falls into the group of diffuse porous angiosperms, just like species of *Fagus*, *Acer Populus*(Poplar/Aspen) and many more. However, there are also ring porous angiosperm genera like *Quercus* (oak), *Castanea* (chestnut), *Fraxinus* (ash), *Ulmus* (elm) and *Juglans* (walnut) which feature a completely different xylem structure with much fewer but far bigger pores. The xylem structure of another big group of trees, the gymnosperms (mainly conifers) also differs markedly from diffuse porous trees, even though the results of Čermák et al. (2008) do not suggest completely different radial velocity patterns for the coniferous *Pinus sylvestris*. Nevertheless, it is very likely that anatomically differing kinds of xylem will behave differently in terms of water transport (Steppe and Lemeur, 2007). Therefore we would highly recommend to include ring porous trees and conifers into future investigations of xylem tracer transport velocity distributions.

## 4.3 Sectoral distribution of tracer transport velocities

We did not systematically look into the sectoral distribution of tracer transport velocities. According to observations of Kozlowski and Winget (1963) and Waisel et al. (1972), water transport through tree xylem rarely follows straight trajectories. Spiraling patterns are very common and the best way to investigate such patterns are locally limited dye tracer injections. Such injections are not suited to assess radial tracer transport velocity distributions as in our study. We did observe some spiraling patterns for the cut-surfaces #15 – #17, indicating that the fastest tracer transport happened along some preferential paths, but these paths seem to be so twisted, that it is not possible to attribute faster transport velocities to a certain sector of the stem.

## 4.4 Velocity distributions captured by different sampling approaches

Considering the differences between the waters that are sampled by the various discussed sampling approaches, it is reassuring to see that there are multiple approaches which do not dramatically deviate from the xylem water flow $Q$, which is probably closest to what we actually intend to measure. Even though in most real world applications xylem core samples are more likely to resemble the seemingly unproblematic *Short core* case (where the core does not reach the center of the stem), the *Deep core* case might apply to cores taken from thinner stems or branches. Our virtual experiments indicate, that in such cases it might be much better to sample entire stem or branch segments (similar to the $V_{total}$ case) or to come up with other solutions that prevent a disproportional contribution of the inner xylem, which might lead to a sampling bias towards older xylem water.

We also have to acknowledge that the immobile water fraction of 10% within conducting xylem depths (related to cell bound water which may be released by aggressive extraction techniques like CVD) in our virtual comparison experiment was just an arbitrary assumption, which might be worthwhile to scrutinize in real world experiments. Real world comparisons of different approaches are rare (Volkmann et al., 2016b; Mennekes et al., 2021; Zuecco et al., 2022) and so far have not been conducted with sufficient temporal resolution to compare temporal dynamics of different approaches. On top of that, other issues related to the differences between sampling approaches themselves, as discussed by Millar et al. (2022), may cause bigger discrepancies than mere differences in the temporal dynamics of the captured water pools.

## 4.5 Implications for stable water isotope aided investigations of RWU

Our dye tracer experiment showed that water transport within the xylem is happening at a large variance in the radial distribution of velocities. This means, that the propagation of isotopic signatures through the xylem cannot always be properly represented by a simple piston flow model like previously attempted by De Deurwaerder et al. (2020); Knighton et al. (2020) and Magh et al. (2020). Moreover, since even within the conducting sapwood only a certain fraction of the xylem is contributing to the water transport, the mean tracer transport velocity can be expected to be higher than the mean sap flux velocity (in our case it was around three times as fast). Maximum tracer transport velocities are even higher (in our case five times as fast as the mean sap flux velocity). As the xylem water is transported at different velocities, abrupt changes of $\delta_{xyl}$ at the base of the stem will be blurred along their way towards the crown. This means, that $\delta_{xyl}$ measurements close to the stem base will be more similar to actual $\delta_{RWU}$. But since RWU is happening at various points throughout the root system, even $\delta_{xyl}$ measurements right at the stem base are not necessarily reflecting current $\delta_{RWU}$ (Seeger and Weiler, 2021). Instead, $\delta_{RWU}$ signals are blurred by a combination of the the TVDs within the root xylem and different flow path lengths (due to the spatial organization of the root system) before they even reach the stem base.

Current review studies (Sprenger et al., 2019; Dubbert et al., 2023) acknowledge a certain relevance of plant internal water travel times but at they also state that the current understanding of them is limited. We think, that investigations of RWU that use $\delta_{xyl}$ measurements should aim for a correct representation of the relationship between $\delta_{RWU}$ and $\delta_{xyl}$. TVDs within the xylem are an essential component of this relationship. They should be studied systematically in order to improve process based plant water uptake modeling.

## 5 Conclusions

In this paper, we presented an inexpensive approach to assess xylem internal water transport velocity distributions and we observed a wide range of transport velocities within the xylem with a mean and maximum velocity of about 300% and 500% of the volumetric water flux, respectively. Velocity distributions at certain xylem depths tended to be normal distributed around
365 mean values that peaked shortly after the outer xylem and decreased towards the inner xylem.

By means of a theoretical assessment of the water transport velocity distributions associated to different xylem water sampling approaches, we found that most approaches sample water with specific underlying velocity distributions similar to that of the actual throughflow of xylem water. A notable exception would be comparatively deep cores from thin stems, which might capture disproportionately high fractions of very slow and immobile water.

The common assumption of an equivalency between $\delta_{RWU}$ and $\delta_{xyl}$ is a simplification. Water that is taken up at a specific point in time cannot be sampled from the xylem - neither at that specific point in time nor with a certain delay. Measurements of $\delta_{xyl}$ will always be composed of a mixture of waters taken up at different points in the past. In case of low $\delta_{RWU}$ variability or comparatively high transport velocities, the effects described in this paper may be negligible, but especially for the correct interpretation of high-frequency $\delta_{xyl}$ observations such water transport velocity distributions should be kept in mind.

*Code and data availability.* Dye tracer experiment sample data

The supplement of this paper contains the raw images that were obtained within the presented dye tracer experiment as well as a collection of R-Scripts that can be used to reproduce the processing chain that lead us to produce Fig. 4 and Fig. 5 of this paper.

*Author contributions.* SS planned, executed and analysed the dye tracer experiment, conducted all computations and wrote the first draft of
380 the manuscript. MW participated in the discussion and helped to complete the manuscript.

*Competing interests.* The authors declare no competing interests.

*Acknowledgements.* We would like to thank Annette Bösmeier for her assistance during the dye-tracer experiment and Natalie Orlowski for her constructive review of the manuscript.

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
