# Peer review of "Dye tracer aided investigation of xylem water transport velocity distributions"

_EGUsphere, 2022_

## Author Comment (AC2)

**Authors' responses to the comments of reviewer#1, James Knighton**

>> *Major comment: the proposed methodology hasn't been validated with a parallel established method.*

Direct validation with a parallel established method surely would have been desirable, but would have required more resources (technical equipment, sample analysis, time) than were available during the making of the presented study. However, we do think that the results of other studies that reported sap flow velocity profiles or breakthrough curves of isotopic labelling experiments are in good agreement with our findings.
* * *
>> *There is some discussion of how these results compare to radial variations in xylem water velocity measured with sapflux techniques; however, these prior measurements were made at different heights, depths into the xylem, and on different species.*

We are well aware that our selected specimen was far from being very representative. But this study was not supposed to deliver final representative values for more relevant tree species, but rather to introduce a methodology that might be suited to investigate sap flow velocity distributions for a wide range of trees of different species and ages. In the revised version of the manuscript we will make sure to highlight that the obtained values should not be generalized.
* * *
>> *I would also guess that this underestimate is what causes the authors to calculate that a substantial proportion of outer xylem tissue does not conduct water (Fig. 6c-d). It may be the case, but this type of finding would impact the conclusions of many prior studies and should therefore be carefully validated.*

We think that is a misunderstanding: The unstained areas in the outer xylem of cut surface #15 (as shown in Fig. 2) do not imply that any part of the outer xylem is not conducting water. They merely imply that some water flowing in the outer xylem was not fast enough to reach cut surface #15 within the time of the experiment. In fact, our results in Fig. 5b-c show that all transport velocities within the outer xylem were greater than 0. What our dye tracer experiment could show, is that the water in the outermost xylem is not moving the fasted. As already mentioned in the discussion of the manuscript, similar findings were reported by studies based on sap flux measurements on different species (see Luttschwager2007, Cermak2008 and Gebauer2008).

The immobile water fractions shown in Fig. 6c-d are actually a combination of two kinds of "immobile" water: first: the 10% of cell bound water that is assumed throughout the entire xylem and second: the "free" water in the inner xylem that simply is not moving. Our proposed dye tracer method would certainly not be suited to quantify the fraction of cell bound water. Even when 100% of a xylem section would appear stained from a dye tracer, we still would have to expect that a certain fraction the water (i.e. the cell bound water) is actually not moving at all. However, the work of Fabiani2022 clearly suggests that there is some exchange between the water in the transpiration stream and less mobile xylem water. We will make sure to include this into the discussion.
* * *
[Figure]

*Extra-Figure 1: Original images of the last seven cut surfaces on top of three rows that used different NDBI thresholds to detect blue stained areas.*

*>> This is a concern to me particularly because the proposed methodology required subjective visual tuning of the NDBI classification of dye versus no-dye in each cross section. This visual tuning seems to have resulted in an underestimate of the dye-stained area (Fig. 2)*

After your hint to the obvious underestimation in Fig. 2 we reran our scripts and actually found that an NDBI threshold of -0.11 produces better results. However even with that threshold we did not manage to properly detect the visibly blue stained areas in cut surface #18. We have created Extra-Figure 1 that shows the last seven cut surfaces together with their digitized versions for three different NDBI thresholds: -0.05 (as in the original manuscript) -0.11 (optimum) and -0.25 (obvious overshoot). Interestingly, an NDBI threshold of -0.25 seems to be suited to detect the faintly blue stained areas of cut surface #18, but on other images this threshold leads to clear misclassifications with too many false positives. We do agree that the automated detection of blue stained area fractions has potential for improvement. We already discussed many of the possible improvements that could be made to obtain better images and we will add your suggestion of reference swatches to the discussion.
* * *
*>> I would also guess that this underestimate is what causes the authors to calculate that a substantial proportion of outer xylem tissue does not conduct water (Fig. 6c-d). It may be the case, but this type of finding would impact the conclusions of many prior studies and should therefore be carefully validated.*

We think that is a misunderstanding: The unstained areas in the outer xylem of cut surface #15 (as shown in Fig. 2) do not imply that any part of the outer xylem is not conducting water. They merely imply that some water flowing in the outer xylem was not fast enough to reach cut surface #15

within the time of the experiment. In fact, our results in Fig. 5b-c show that all transport velocities within the outer xylem were greater than 0. What our dye tracer experiment could show, is that the water in the outermost xylem is not moving the fasted. As already mentioned in the discussion of the manuscript, similar findings were reported by studies based on sap flux measurements on different species (see Luttschwager2007, Cermak2008 and Gebauer2008).

The immobile water fractions shown in Fig. 6c-d are actually a combination of two kinds of "immobile" water: first: the 10% of cell bound water that is assumed throughout the entire xylem and second: the "free" water in the inner xylem that simply is not moving. Our proposed dye tracer method would certainly not be suited to quantify the fraction of cell bound water. Even when 100% of a xylem section would appear stained from a dye tracer, we still would have to expect that a certain fraction the water (i.e. the cell bound water) is actually not moving at all. However, the work of Fabiani2022 clearly suggests that there is some exchange between the water in the transpiration stream and less mobile xylem water. We will make sure to include this into the discussion.
* * *
>> *The ideal resolution to this comment would be to provide a validation or comparison of the technique. I would imagine that this experiment could accommodate sap-flux sensors installed along the sample and/or xylem water isotopic sampling (if labeled and dyed water were used at the boundary condition). I recognize that this would be a substantial amount of work. At the very least, I strongly caution against using the results of this preliminary experiment to guide field sampling and model development and think the results and conclusions could be tempered a bit more than they already have been.*

Within the revised version of the manuscript we will make sure to clarify that the results of this study should not directly be the basis for decisions regarding field sampling or model development. The presented study could be a starting point to devise more refined experiments in order to investigate the distribution of flow velocities within the xylem.

Regarding your suggestion for validation with other approaches, we would argue that our findings fit very well to already published distributed sap flow measurements (Luttschwager2007, Cermak2008 and Gebauer2008), even though those measurements may have been made at different species and at different spatial dimensions. The dye tracer approach may even be an option for smaller plant compartments, that could hardly be investigated by distributed sap flow measurements. The idea of comparing a stable water isotopic label with our dye tracer was already mentioned in the discussion part of the manuscript (lines 263-265), but this approach would not be suited to evaluate the spatial resolution that can be achieved with the dye tracer approach. At least for tracing the water flux through soil, a recent study by Llorens2022 showed that isotopic labeling and brilliant blue labelling do lead to pretty similar results.
* * *
>> *Section 1: I appreciate the focus on process-based modeling of root to xylem conduit time lags. I would also point out that research using storage selection (SAS) functions have been presented as a parallel way of dealing with this problem. These models don't need to explicitly consider where water spent time (i.e., inside the plant versus in the soil). SAS functions can numerically represent complex age-distributions of water in xylem. I understand that this isn't the primary focus of the paper, but this comes to mind as I read the argument the authors are making. Here are several examples...*

We are aware of SAS functions, but within our introduction we tried to trace the path from traditional approaches towards more process based approaches that enable a more realistic

representation of xylem water transport. From that perspective we would prefer to keep the introduction short and not include additional references towards a purely statistical/parametric approach (i.e. SAS functions) that plays no role for the further content of the study.
* * *
>> *Line 49: The process-based modeling papers described by the authors all shared a similar challenge in that they were attempting to simultaneously simulate the both the soil water balance (the root boundary condition) and xylem water isotopic observations. Some of the model error in representing xylem water was likely error that cascaded down from an imperfect representation of the soil water, and not only an imperfect representation of xylem water transport. Smith et al (2022) demonstrated this directly. This might be worth mentioning.*

At least for the study of Seeger2021 the soil water used for the computations did not come from a model, but from interpolated in-situ measurements of soil moisture and soil water isotopes. Therefore, I do not think that this point applies here. The observed discrepancies in that work clearly were caused by a lack of a proper representation of water storage and transport within the xylem.
* * *
*Line 55: This transition in text is kind of abrupt. The text goes from advances in modeling the soil-plant system to measurement techniques. Would this paragraph make more sense moved down to be the first paragraph of Section 1.2?*

We agree that the transition is abrupt. Instead of moving it to the first paragraph of section 1.2, we would place them into a new subsection about tracers (dyes and isotopes) that will open section 1 of the revised manuscript.
* * *
>> *Line 75: A definition of "mobile" would be useful here. Is "mobile" water extracted under a certain pressure? Many researchers remove heartwood from tree core samples prior to isotopic analysis to sample only the "mobile" water; however, recent studies have suggested that heartwood is "mobile" to a certain degree. I think the field at large needs a better definition for "mobile." For example, see Fabiani et al (2022).*

As already mentioned above, we are actually dealing with two kinds of immobile water (cell bound water and free water that is not moving). We will make sure to clearly define what we are talking about at this point.
* * *
>> *Line 87: Possibly no one has done this with stable isotopic techniques or dye tracers, but there are many studies where the radial variations in xylem velocity have been observed (Ford et al 2004). Maybe I'm not understanding what the authors are suggesting, but this seems a like a new method of measuring something that has been measured many times before.*

The point is that sap flux velocities and water transport velocities within the xylem are not identical - in our case the water transport is on average three times (and at a maximum five times) faster than the mean sap flux velocity. Furthermore, sap flux sensors do give one value per depth class (usually not finer than 5 mm) while the dye tracer approach allows for a much higher spatial resolution. Additionally, the dye tracer approach returns whole velocity distributions for each depth class. In order to achieve the same with sap flux sensors, unreasonable amounts of sensors would be needed.
* * *
*>>Line 107: How long did this dye breakthrough require?*

42 minutes. This information is given right at the beginning of the results section (line 183).
* * *
*>> Figure 6c-e: This is implying that some shallow xylem tissue existed that was not transporting water. It seems more likely that this was because the NDBI algorithm was missing areas where dye was showing up weakly. You can see that this is likely the case on the left-hand side of Figure 2a and 2c. There is a clear blue-green coloring along the left of the cross section that was not identified by the algorithm and (I'm assuming) coded as "immobile." The authors also point out that the blue dye leaked from the last cross section (mobile) but did not stain this cross section (not identified as mobile). I would guess that the reported "immobile" water percentages are more related to methodological limitations than evidence that immobile water is being sampled in short cores.*

I hope that part was already answered above. Short repetition: Undyed areas further down the flow path length merely imply that the water transport was not fast enough to reach this point within the time of the experiment – it does not mean that the water is not flowing. The immobile water that can be expected to be sampled as part of short cores is cell bound water, which might be released if an aggressive extraction method like CVD is used.
* * *
*>> Line 225: I'm not sure if it would work with this the methodology, but we often apply mineral oil to increase visibility when counting rings on tree cores. It makes the wood grain much more distinct. This might be an inexpensive way to quickly improve image quality.*

*>>Line 230: You could also include a reference tile or swatch next to each stem cross section for image post-processing to a visual standard.*

Thank you for these suggestions, we will include these suggestions to the discussion of the method.
* * *
*>> Line 284: The wording here is a bit awkward: "a lot more, very common trees."*

This will be rephrased in the revised manuscript.
* * *
*>> Line 295 - 296: This recommendation seems like an overreach given that this methodology hasn't been validated (see major comments).*

Actually, for mature trees the deep core case will rarely ever apply. But we will rewrite that part.
* * *
*>> Line 304: "wide range of velocities" maybe isn't the most accurate wording. As discussed in section 4.2 this study shouldn't be taken as a measurement of velocity because of the unusual boundary conditions. Maybe instead: "a large variance in the radial distribution of velocities" which avoids talking about the actual velocities.*

This will be rephrased in the revised manuscript.
* * *
*>> Line 305: Minor comment here, but Knighton et al (2020) investigated both piston flow and a fully-mixed reservoir, which was a little more complicated than piston flow.*

We have mentioned the fully-mixed reservoir of Knighton2020 in section 1.1 of the introduction. At that point, we would argue that a from a physical perspective the fully-mixed reservoir seems less plausible than a piston flow representation. Therefore, we do not think that the manuscript would gain anything by mentioning the fully-mixed reservoir at that point.
* * *
*>> Line 305: There is also a possible difference in the temporal scales between that modeling experiment and this study presented in the manuscript. I'm assuming this study occurred over a time period of less than one working day (<8 hours). If a boundary condition is introduced and held constant for a sufficient period of time (e.g., rainfall onto soils followed by a period of no rain for a week) the radial distribution of xylem velocities will stop mattering after the slowest path has reached the sampling height. The variation in flowpath velocities would only be a strong consideration if the water boundary condition at the roots was rapidly changing relative to the transit time between roots to the sampling point, which might not always be a real-world concern.*

We agree that the effect introduced by a distribution of transport velocities may be negligible whenever the considered systems can be considered to be close enough to steady state conditions. They also do not play a role when the ratio of transport distance to velocity is small enough. However, as shown by Seeger2021, a temporal lag between RWU and xylem water measurement as well as non-negligible blurring introduced by different transport velocities do occur at real world uptake conditions (even though artificial labelling can enhance the visibility of such effects). Even if a less complex representation might not do any harm in most cases, there will definitely be occasions where the inclusion of transport velocity distributions can help to achieve a more realistic relationship between the observed signatures in RWU and xylem water.
* * *
*>> Line 316: Something wrong with this reference "?Dubbert"*

Error in the bibtex file was fixed.

**References**

Lüttschwager, D., and Rainer, R.:
"Radial distribution of sap flux density in trunks of a mature beech stand." *Annals of forest science* 64.4 (2007): 431-438.

Čermák, J., Nadezhdina, N., Meiresonne, L., and Ceulemans, R.:
"Scots Pine Root Distribution Derived from Radial Sap Flow Patterns in Stems of Large Leaning Trees" Plant and Soil, 305, 61–75, https://doi.org/10.1007/s11104-007-9433-z, 2008.

---

## Author Comment (AC3)

**Authors' responses to the comments of reviewer#2**

*>>The table and figures are lacking important information. Explanation for abbreviations is widely missing. All figures need to be explained better in the figure legends, not just in the text. A lot of information can be only found in the text, see e.g. information in lines L189, L209-210. The legend of Fig. 4b for instance is not referring at all to the flow path length (x-axis of the figure).*

We will make sure to add the missing information in the revised version of the manuscript.
* * *
*>>It would be great if the figures would also use/refer to the abbreviations from the equations for better understanding.*

We will incorporate these suggestions into the revised version.
* * *
*>>The connection between 1 Introduction, 1.1 and 1.2 should be made clearer. The relevance of 1.1 and 1.2 only get clearer when the objectives (1.3) are listed. Some sentences at the end of section 1 should introduce 1.1 and 1.2.*

We will rework the mentioned sections for the revised version of the manuscript.
* * *
*>>How is the variation within a xylem class? Looking at Figure 4, also the trunk side of the sampled tree mattered. Figure 3a assumes no large variation within the xylem class? This point of variation within a xylem depth class should be also discussed.*

The variation within xylem depth classes is depicted in Fig. 5b and 5c and it is described in lines 201-203. The trunk side brings up a complex issue which was neglected within the initial manuscript. We have found two studies (Kozlowski & Winget 1963 and Waisel et al. 1972) which have investigated how a dye tracer injected to a certain sector of the stem is propagated throughout the xylem. A straight sectoral propagation of the dye signal seems to be far less common than spiraling and increasingly spreading propagation patterns. This means that a sectoral analysis of the apparent transport velocities derived from dye tracing experiments like the one presented in this study could be problematic and lead to misinterpretations. This issue will be included to the discussion of the revised version of the manuscript.
The velocity distributions that are used for Figure 3a assume sectorally homogeneous velocity distributions for each depth class that resemble a normal distribution, as described in lines 160-165, which are idealized, but similar to the observed distributions which are shown in Fig 5b.
* * *
*>> Will you share your codes with the community?*

After some cleaning and commenting, the processing scripts with some example input files will be made available to the community.

**Responses to (some of) the line by line comments**
*>>L3: you only study the plant water uptake/transport starting from the stem base*

That is correct - even though it stands to reason that water transport through the roots also cannot be instantaneous, it is probably better to focus the abstract more on what was investigated within this study.
* * *
*>>L79: CVD abbreviation not introduced*

We will add that abbreviation to line 69, where it should have been introduced.
* * *
*>> L82: the borehole method likely only sees the last few cm of the borehole before the airstream is carried out of the stream*

We will add the distinction between deep and short cores at that point and make sure that the borehole method will be introduced as an example for the short core case.
* * *
*>> L115: Figure 2, could you add a scale as a reference?*

*>> L118: how big is one pixel?*

We will add a scale to Fig.2 and we will also add the approximate size of one pixel.
* * *
*>> L120: how were the classes defined exactly? Be more precise. Why six classes? I would already add "(Fig.2c)" here*

The classes where defined as six equally spaced distances classes from the xylem edge. The number of classes was chosen as a compromise between spatial resolution and signal to noise ratio. We will make sure to add this information to the revised version of the manuscript.
* * *
*>> L124: might be nicer to write „Eq. 1" etc. As you also reference to them later.*

We see the point, but since we used the official Latex-Template provided by Copernicus, we are not sure whether it is up to us to deviate from the official layout. Maybe the editor can answer that question?
* * *
*>> L133: „required velocities $u_n$ to reach"*

We will add this to the revised manuscript.
* * *
*>> L152: add more information „does not matter because…"*

What we mean is that the hypothetical stem segment has an arbitrary size. Its length does not matter for the consideration of the velocity distribution at any cross section. We will rephrase the introduction of this section…
* * *
*>> Figure 3; four different kinds of tiny circles? do you mean shapes/colour groups?*

We will rewrite the caption of Figure 3 to make it clearer.
* * *
>> *L160-165: explain / reference the choice for innermost=0, outermost decreases again*

We based this on the observed pattern of our dye tracer model. Similar velocity distributions, based on distributed sap flow measurements, have been observed by Lüttschwager & Rainer 2007 and Čermák et al. 2008. We will add these references.
* * *
>> *L170: "any study"*

>> *L172: „different xylem sampling approaches will capture"*

>> *L174. intercomparison, inter-comparison reads nicer*

We will gladly take these comments into account for the revision of the manuscript.
* * *
>> *L185: how much was the thicker end of the stem?*

The thicker end of the stem had a diameter of 38 mm.
* * *
>> *L186: you could add „Eq. 6"*

We will add that reference.
* * *
>> *L194: „virtually" is confusing here as this is the actual experiment*

We meant this in the sense of "practically", "almost" or "nearly". We will rephrase that part to avoid confusion in the revised manuscript.
* * *
>> *L199: equation 2.1.2??*

That was supposed to be "Eq.6" – misplaced a label within the LaTeX source file…
* * *
>> *Fig.5 legend. „per xylem depth class", (c) is actually the xylem radius (be more consistent)*

Each xylem radius is falling into one of the depth classes. We tended to use the two terms interchangeably throughout the whole paper. Upon your comment we see how this can lead to confusion, especially if a single Figure is contemplated with no further context than its caption. We will make sure to be more consistent in the revised version of the manuscript.
* * *
>> *Fig.6 Frequency: y-axis labels*

The absolute numbers on those y-axes do not matter, since they completely depend on the respective resolution of the number of points "sampled" along the x axis. That's why we chose to

omit specific values on the y-axes. Would you prefer to see some arbitrary numbers on the y-axes or would it be enough to add this information to the caption?
* * *
*>> L247-251: maybe make two sentences out of this very long one*

We will split that sentence in the revised version of the manuscript.
* * *
*>> L255-L261: mention that velocities can be highly species-specific and size of tree etc.*

We will add that information to the revised version of the manuscript.
* * *
*>> L258: wasn't this even in three heights?*

Yes, it was, but the intermediate observation (at breast height) did not add much to the overall results. For the sake of completeness, we will describe the actual instead of the "effective" setup.
* * *
*>> L265: did you sample your stem for isotopic analysis? so that you could compare it directly?*

No, we did not. Obtaining a whole breakthrough curve from drip samples would have required more liquid water isotopic analyses than were available within this low budget experiment.
* * *
*>> L266-274: these are all different species, might be worth to emphasize this more*

We already clearly stated the species for each of the cited studies, but we can of course additionally emphasize, that they were not the same as in our experiment.

*>> L285: „more conductive pores" than? be more precise*

We will rephrase this sentence in the revised version of the manuscript.
* * *
*>> L270: start new sentence after "… beech)"*

*>> L284: phrasing*

*>> L287: „trees. It is"*

*>> L288: behave differently*

*>> L294: overrepresentation, over-representation reads nicer*

*>> L316: phrasing*

*>> L316: ?Dubbert, delete"?"*

These comments will be considered during the revision of the manuscript.
* * *
**References**

Kozlowski, T.T. and Winget, C.H.:
"Patterns of Water Movement in Forest Trees"
Botanical Gazette, Vol. 124, No. 4 (Jun., 1963)

Waisel, Y., Liphschitz, N. and Kuller, Z.:
"Patterns of Water Movement in Trees and Shrubs"
Ecology, Vol. 53, No. 3 (May, 1972)

Lüttschwager, D., and Rainer, R.:
"Radial distribution of sap flux density in trunks of a mature beech stand." *Annals of forest science*
64.4 (2007): 431-438.

Čermák, J., Nadezhdina, N., Meiresonne, L., and Ceulemans, R.:
"Scots Pine Root Distribution Derived from Radial Sap Flow Patterns in Stems of Large Leaning Trees"
Plant and Soil, 305, 61–75, https://doi.org/10.1007/s11104-007-9433-z, 2008.

---

## Author Response (AR1)

**Author's response for the minor revisions of the manuscript "Dye tracer aided investigation of xylem water transport velocity distributions"**

Generally, we would like to refer to the point by point replies to the reviewer comments postet in the open review process as AC2 and AC3. Within this document we will not repeat every point that was made within AC2 and AC3. We will rather focus on documenting bigger changes that were made to the manuscript during the revision process.

In section 1.1, we moved the paragraph describing dye tracer approches to investigate plant water transport from the end to the start.

Since Reviewer #1 noted a very abrupt transition from section 1.1 to section 1.2, we added an introductory paragraph to section 1.2 and hope that this improves the reading experience.

We changed Fig. 2 to include a scale and legends.

In the Results chapter, we added some information on the NDBI threshold and the spatial resolution achieved within our analyses.

For our description of the results, we changed the naming order of the xylem depth classes (now depth class number 1 refers to the outer xylem and depth class number 6 to the inner xylem, which is hopefully more intuitive than the previously used naming scheme).

We changed Fig. 4 to depict bigger classified images of the stained surfaces (in Fig.4a) and included a hopefully more intuitive legend for the xylem depth classes.

We added a similar depth class legend to Fig. 5 and made some minor adjustments, most importantly labeling the axes with full words instead of cryptic things like "u/q [-]".

We also adapted Fig.6 to include visual representations of the sampled parts of the xylem cross section. This also should make it more easy to relate the colored lines in Fig.6 to the respective depth classes.

In the discussion of Sec. 4.2 we included a paragraph that highlights the difference between radial velocity distributions inferred by sap-flux sensors and by dye tracers.

We added section 4.3 to discuss the sectoral distribution of tracer transport velocities and to justify, why we did not assess them in detail.

We also invested quite some effort to compile a hopefully comprehensible collection of scripts and data that should enable others to reproduce the results presented in this study.